# Impact of the Loading Conditions and the Building Directions on the Mechanical Behavior of Biomedical *β*-Titanium Alloy Produced In Situ by Laser-Based Powder Bed Fusion

**DOI:** 10.3390/ma15020509

**Published:** 2022-01-10

**Authors:** Housseme Ben Boubaker, Pascal Laheurte, Gael Le Coz, Seyyed-Saeid Biriaie, Paul Didier, Paul Lohmuller, Abdelhadi Moufki

**Affiliations:** 1Universite de Lorraine, CNRS, LEM3, Arts et Metiers ParisTech, 57070 Metz, France; gael.lecoz@univ-lorraine.fr (G.L.C.); seyyed-saeid.biriaie@univ-lorraine.fr (S.-S.B.); paul.didier@univ-lorraine.fr (P.D.); paul.lohmuller@univ-lorraine.fr (P.L.); abdelhadi.moufki@univ-lorraine.fr (A.M.); 2Universite de Lorraine, CNRS, LEM3, IMT, GIP InSIC, 88100 Saint Die des Vosges, France

**Keywords:** Ti-Nb alloy, additive manufacturing, selective laser melting, characterization, thermomechanical behavior

## Abstract

In order to simulate micromachining of Ti-Nb medical devices produced in situ by selective laser melting, it is necessary to use constitutive models that allow one to reproduce accurately the material behavior under extreme loading conditions. The identification of these models is often performed using experimental tension or compression data. In this work, compression tests are conducted to investigate the impact of the loading conditions and the laser-based powder bed fusion (LB-PBF) building directions on the mechanical behavior of β-Ti42Nb alloy. Compression tests are performed under two strain rates (1 s${}^{-1}$ and 10 s${}^{-1}$) and four temperatures (298 K, 673 K, 873 K and 1073 K). Two LB-PBF building directions are used for manufacturing the compression specimens. Therefore, different metallographic analyses (i.e., optical microscopy (OM), scanning electron microscopy (SEM), energy-dispersive X-ray (EDX), electron backscatter diffraction (EBSD) and X-ray diffraction) have been carried out on the deformed specimens to gain insight into the impact of the loading conditions on microstucture alterations. According to the results, whatever the loading conditions are, specimens manufactured with a building direction of 45${}^{\circ }$ exhibit higher flow stress than those produced with a building direction of 90${}^{\circ }$, highlighting the anisotropy of the as-LB-PBFed alloy. Additionally, the deformed alloy exhibits at room temperature a yielding strength of 1180 ± 40 MPa and a micro-hardness of 310 ± 7 HV${}_{0.1}$. Experimental observations demonstrated two strain localization modes: a highly deformed region corresponding to the localization of the plastic deformation in the central region of specimens and perpendicular to the compression direction and an adiabatic shear band oriented with an angle of ±45 with respect to same direction.

## 1. Introduction

Owing to their good corrosion resistance [1,2] and their excellent mechanical properties, such as high strength and low Young’s modulus [3], Ti-Nb titanium alloys have attracted a lot of attention to be candidates for the fabrication of the next generation of implants [4,5,6]. β-Ti-Nb alloys have a relatively low Young’s modulus [7,8], which is due to the niobium (Nb) β-stabilizing element that retains the β-phase at an ambient temperature. For the biomedical applications, Ti-Nb alloys have also shown excellent bio-compatibility properties [9,10]. Moreover, Ti-Nb alloys reduce the undesired stress shielding effect caused by the mechanical incompatibility between the implant and the bone [11,12,13,14].

Recently, the emergence of additive manufacturing technologies such as laser-based powder bed fusion [15,16,17] and electron beam melting [18,19,20] techniques has accelerated the use of β Ti-Nb alloys for the fabrication of implants [21]. Additive manufacturing technologies allow us to generate complicated shapes and eliminate the unnecessary processes, which reduces the fabrication cost [22,23]. In addition, excellent thermomechanical properties can be obtained after LB-PBF additive manufacturing of Ti-Nb alloys [24]. By adopting the appropriate powder size, LB-PBF process parameters and thermomechanical heat treatments, mechanical properties of parts produced using the LB-PBF process can be much more improved than those of the plasma sintering, pressure-less sintering and forged parts [25,26,27,28].

The differences in density, powder size, powder-to-laser absorptivity and melting temperature of Ti and Nb elements lead to some limitations for obtaining a homogeneous as-build microstructure of Ti-Nb alloy. Ti-Nb microstructures almost show undissolved niobium particles and pores. For improving the microstructure homogeneity, several studies have been conducted using different processing parameters. Santos et al. [29] studied the effect of LB-PBF parameters on Ti35Nb (in wt.%) microstructure. According to the results, the LB-PBF parameters greatly affected the porosity amount and the desired mechanical properties of the alloy, such as the elastic modulus and compressive strength. Wang et al. [30] investigated the microstructure and the mechanical behavior of Ti35Nb alloy (wt.%) produced by laser-based powder bed fusion on elemental powder mixture. The results showed that the Ti35Nb microstructure was composed of dendrite β phase and undissolved Nb powders. It was demonstrated that the heat treatments applied to the LB-PBF-parts improved their chemical homogeneities through dissolving most of the un-melted Nb particles. The effect of the chemical composition on the as-LB-PBFed microstructures of Ti-Nb alloys was investigated by Wang et al. [26]. For this purpose, three Ti-Nb alloys were considered, which contained three amounts of Nb powders: Ti26Nb, Ti40Nb and Ti62Nb (in wt.%), respectively. According to the experimental results, an increase in the Nb amount of the powder reduces the martensitic transformation and the the grain size of the alloy and increases the β phase amount. The tensile test results illustrated that the lowest Young’s modulus and best strength/ductility ratio were obtained for Ti40Nb alloy. Similar trends have been found by Yılmaz et al. [31]. For Ti-xNb alloys elaborated by the laser engineered net shaping method, Damian et al. [32] demonstrated that the lowest Young’s modulus was observed for Ti33Nb alloy compared to the Ti25Nb, Ti38Nb and Ti46Nb alloys.

Based on literature review, it appears that great attention has been paid to the impact of the LB-PBF parameters on microstructures and mechanical properties of β Ti-Nb alloys. However, no paper has figured out the effect of the temperature and the strain rates on the deformation behavior of β Ti-Nb alloys. Additionally, little attention has been paid to the impact of the material anisotropy on the mechanical behavior of Ti-β alloys produced by selective laser melting. However, because of the poor surface integrity (i.e., residual stress and surface roughness), Ti-Nb β Ti-Nb alloys as implantable medical devices need to be machined in order to obtain a functional surface finish. The machining process generally induces severe loading conditions such as high strain rates and high temperatures, which can lead to some microstructure changes. The literature review showed a lack of studies about the effect of loading conditions on the thermomechanical response and the deformation behavior of this alloy.

The current work aims, therefore, to investigate the impact of the temperatures and the strain rates on the thermomechanical behavior of the Ti42Nb alloy produced with different LB-PBF building directions. Compression tests are conducted at two strain rates (1 and 10 s${}^{-1}$) and temperatures ranging from 298 to 1073 K. In addition, two as-build LB-PBF building directions are considered for investigating the material anisotropy effect. Therefore, metallographic investigations are performed on the deformed specimens in order to understand the effect of microstructural transformations on the mechanical behavior. This paper is organized as follows. In the first section, the experimental procedure to perform the uniaxial compression tests is detailed. The impact of the loading conditions on the Ti42Nb deformation behavior is presented in the second section. Finally, the relationship between microstructural transformations and thermomechanical behavior is discussed in the third section.

## 2. Materials and Methods

Chemical analyses are carried out on the powders as well as the specimens produced. These analyses provide the concentration of the hydrogen, nitrogen and oxygen elements in the specimens. The corresponding chemical composition of the Ti42Nb titanium alloy employed in this study is given in Table 1.

In this work, the behavior of the as-LB-PBFed Ti42Nb alloy manufactured from the elemental mixture of spherical Ti and Nb powders is investigated. Particle size distributions are measured by laser diffraction. Particle size analyses revealed that Ti particles sizes ranged from $D_{10}$ = 26 μm to $D_{90}$ = 118 μm with $D_{50}$ = 62 μm while the size of Nb particles is $D_{10}$ = 12 μm, $D_{50}$ = 44 μm and $D_{90}$ = 90 μm. SEM observations illustrated in Figure 1 reveal the presence of a small nuumber of satellites.

For manufacturing the compression specimens, a LB-PBF 280 machine with a maximum power of 400 W is used. The laser spot diameter is 80 μm. The specimens are fabricated using optimized LB-PBF parameters investigated in the experiments of Fischer et al. [33]. The laser power, the scanning speed, the scanning spacing and the layer thickness are set to be 300 W, 300 mm/s, 60 μm and 30 μm, respectively. A cross-hatching scanning strategy with 67${}^{\circ }$ rotation is adopted to avoid the formation of interconnected porosity [34]. The building chamber is protected by high purity argon gas aimed at avoiding oxidation during the additive manufacturing process.

In this work, the specimens are produced with two as-build directions. The first batch of specimens is produced with a building direction parallel to the Z- axes of the LB-PBF machine. However, a building direction rotated by 45${}^{\circ }$ with respect to the Z-axes is selected for manufacturing the second batch of specimens. In the upcoming sections, these two building directions allow us to generate vertical, and 45${}^{\circ }$-inclined compression specimens are named BD-90${}^{\circ }$ and BD-45${}^{\circ }$, respectively.

Compression tests are performed using a Gleeble 3500 machine. The trials are performed under vacuum condition to avoid oxidation. The cylindrical specimens are first produced by laser-based powder bed fusion and then machined in order to obtain the desired dimensions (i.e., 6 mm in diameter and 9 mm in height) before the experiments. During the tests, the elongation is measured using an axial extensometer. The temperature *T* is controlled with a K-type thermocouple. Meanwhile, in the compression tests, two graphite sheets are placed between the anvils and the two faces of specimens (i.e., upper and lower faces) to reduce the friction and to favor the electrical conduction. The deformation is performed with a constant nominal strain rate of 1 and 10 s${}^{-1}$ up to the fracture of the specimens or an axial strain of 50%. The temperature varies from 298 K to 1073 K. The axial elongation ΔL and the axial force *F* are used for calculating the axial true stress σ and the axial logarithmic strain ε.

Before testing, the specimens are first heated from room temperature to the test temperature with a constant heating rate of 10 K/s. Subsequently, the temperature is held constant during 30 s before the mechanical test in order to limit the temperature gradients. Then, compression tests are performed using a constant nominal strain rate controlled by the Gleeble machine. Finally, a compressed-air jet is used to cool the specimens rapidly from the test temperature to room temperature once the compression test is accomplished.

It is important to highlight that the applied heat cycle does not affect the microstructure (i.e., average size and the presence of phases). The compression trials are repeated three times under identical loading conditions. According to the results, good repeatability is obtained. The flow stress variations between tests prior to fracture are inferior to 4%.

For the observations of the deformed microstructure, the deformed specimens are first cut along the axial cross section. Then, the surfaces containing the compression direction are prepared based on standard procedures, etched by Kroll’s reagent (3% HF and 6% HNO${}_{3}$ in water) and observed with SEM equipped with an EDX and EBSD systems. The phase identification was conducted using XRD with a Cu target in a conventional X-ray tube operating at an accelerating voltage of 40 kV and an electron beam current of 40 mA. The scattering angular (2θ) varied from 30${}^{\circ }$ to 120${}^{\circ }$. Vickers microhardness tests are carried out on the deformed specimens using a ZWICK ZHV1 Microhardness Tester. For the measurements, the load is fixed to 100 g and the holding time is set to 10 s. For obtaining reliable microhardness data, the hardness measurements are performed at 20 points at the central area of each deformed specimens. Microhardness tests are conducted parallel and perpendicular to the compression direction. An ultrasonic method was used to estimate the Young’s modulus of the alloy. The Young’s modulus was deduced by measuring the wave propagating flight time through the material. Two transducers (an emitter and a receiver) were used to propagate longitudinal and transversal waves through the cubic sample. An electrical pulse generator connected to the transducer allows the generation of an acoustic pressure that propagates as a pulse wave through the sample with a 100 Hz pulse repetition frequency. Then, the other transducer receives the wave on its surface and converts it into an electrical signal that can be shown on an oscilloscope. Time sampling of the acquisition is 0.5 ns. The longitudinal and transversal wave velocities are then directly determined. This allowed us to deduce the Young modulus. An EasyTom Nano X-ray micro-tomograph was used to characterize the homogeneity of the as-build samples. This device is equipped with a tungsten filament, which allows us to obtain a maximum resolution of 0.75 μm and an acceleration voltage of 160 kV and thus to acquire high resolution images.

## 3. Results and Discussions

### 3.1. Microstructural Characterization of the as-LB-PBFed Ti42Nb Microstructure

Metallographic investigations were performed to characterize the as-LB-PBFed Ti42Nb microstructure. The scanning electron microscopy observations shown in Figure 2 demonstrate that the as-LB-PBFed Ti42Nb microstructure is composed of large β-grains mainly oriented along the building direction. This microstructure contains non-columnar elongated grain. According to the observations, the grain size is around 80 μm. The grain size is, therefore, greater than the layer thickness. Moreover, the melt pool boundaries are easily identified. For the as-LB-PBFed microstructure, an epitaxially grain growth is observed. The grains grew perpendicular and across the melt pool boundaries towards the top center of each melting track.

As shown in Figure 3b, the XRD pattern of the as-LB-PBF microstructure is only composed of the respective peaks of a single β phase. The lattice parameter calculated from XRD pattern is a=0.3289±0.0001 nm. This value is consistent with the single BCC β phase Ti42Nb alloy parameter [33,35,36]. The peaks of the *α*″ martensitic phase, which are possibly developed during the rapid cooling, are not observed in the XRD spectrum. Similar observations were reported by Wang et al. [26]. This finding proves an accordance with the low Young’s modulus of the as-LB-PBFed Ti42Nb alloy measured by ultrasonic method in this work, around 56.7 ± 2 GPa, which is mainly related to the single β-phase [33].

Figure 3a,b show the presence of un-melted or partially melted Nb particles. Tomography analyses prove that the un-melted Nb particles represent less than 1% of the total amount of the Nb mass in the alloy. Despite that, high laser energy is able to melt the Ti powder with a fusion temperature of 1668 °C; the latter is not sufficient to fully melt the Nb powder with a fusion temperature of 2488 °C. Thus, a high Nb melting point leads to obtain partially or completely not melted Nb powders. According to Li et al. [37], the high ratio of Nb powder could also cause large amounts of unmelted particles.

SEM observations illustrate the presence of spherical and elliptical pores distributed over all the specimens (see Figure 3b). These pores do not have preferential nucleation sites. Some irregular pores are observed around the undissolved Nb particles, which may be induced by the large un-melted Nb powder that acted as stress concentration points and affected locally the heating sources during the solidification process [38]. Similar observations were reported by [26,39].

Finally, electron backscatter diffraction (EBSD) analyses are performed to highlight the existence of preferred crystallographic orientation. The cross section of the as-LB-PBF specimens are therefore analyzed. For BCC materials, the easy-growth directions are $\langle 100\rangle$ since they correspond to the directions of maximum temperature gradient. The easy-growth directions in BCC materials correspond to the building direction in the case of the LB-PBF additive manufacturing processes. The corresponding results are shown in Figure 4 and Figure 5 for specimens produced with building directions of 90${}^{\circ }$ and 45${}^{\circ }$, respectively. According to the results, a preferred crystallographic orientation $\left\{001\right\}\langle 100\rangle$ is observed for the two building directions. EBSD analyses illustrate that most of β grains are oriented parallel to the building direction.

### 3.2. Thermomechanical Behavior of Ti42Nb Alloy

#### 3.2.1. Effect of temperature and strain rate for building direction of 90${}^{\circ }$

Stress–strain curves of the Ti42Nb alloy manufactured with a building direction of BD-90${}^{\circ }$ obtained over a wide range of temperatures from 298 K to 1073 K and strain rates ranging from 1 s ${}^{-1}$ to 10 s${}^{-1}$ are shown in Figure 6. According to the results, the deformation temperature has a great effect on the flow stress and the fracture strain of the alloy fabricated with a building direction of 90${}^{\circ }$. For a fixed strain rate, as the temperature increases, the flow stress decreases and the elongation to fracture increases. The former can be related to the fact that the temperature rise provides sufficient thermal energy to dislocations for overcoming the short-range obstacles [40], while the latter is justified by the impact of the microstructure change, induced by the occurrence of dynamic recovery at intermediate and high temperature, which enhanced the ductility behavior of Ti-β alloys [41]. As shown in Figure 6, the flow stress decreases significantly when the temperature is increased from 298 K to 673 K and drops drastically at 1073 K. At room temperature (T= 298 K), the initial yielding is followed by a continuous softening. However, at intermediate and high temperatures (T= 873 K and T= 1073 K), the alloy revealed a steady state behavior. showing an equilibrium between the strain hardening and the strain softening [42,43]. Similar trends have been observed by [44,45,46]. For those alloys, the strain softening is related to the effect of microstructure alterations, such as dynamic recovery and dynamic recrystallization, which could happen [47,48]. However, at room temperature, the strain softening can be attributed to the shear banding when the material is subjected to high strain rate.

#### 3.2.2. Effect of temperature and strain rate for building direction of 45${}^{\circ }$

Figure 7 shows the stress–strain curves obtained for the Ti42Nb alloy manufactured with the building direction of 45${}^{\circ }$. The compression trials are conducted at temperatures ranging from 298 to 1073 K and different strain rates (1 s${}^{-1}$ and 10 s${}^{-1}$). Ti42Nb produced with a building direction of 45${}^{\circ }$ exhibits a thermoplastic behavior. Therefore, a decrease in the temperature is associated with an increase in the flow stress and a decrease in the fracture strain. Regardless of the loading conditions, Ti42Nb alloy produced with a building direction of 45${}^{\circ }$ exhibits higher flow stress than Ti42Nb manufactured with a building direction of 90${}^{\circ }$. When comparing the stress–strain curves shown in Figure 6 and Figure 7, the effect of the anisotropy becomes more pronounced at high temperatures. When the temperature is increased from 298 K to 1073 K, the flow stress of the specimens produced with a building direction of 45${}^{\circ }$ are 8% and 26% higher than the flow stress of the specimens produced with a building direction of 90${}^{\circ }$. Moreover, specimens manufactured with building direction of 90${}^{\circ }$ demonstrated a better ductility. The mechanical properties of metallic materials are mainly governed by their microstructure. Regarding two key factors, the grain morphology and crystallographic texture can induce anisotropic mechanical properties. Since dislocations could not easily transfer across the grain boundary, the slip path for the sample produced with a building direction of 90${}^{\circ }$ is longer to that of samples fabricated with a building direction of 45${}^{\circ }$. In addition, a contribution of the texture to the anisotropic mechanical behavior cannot be excluded.

### 3.3. Comparison between the Behavior of Ti42Nb Alloys Produced with Building Directions of 90${}^{\circ }$ and 45${}^{\circ }$


In this section, a comparison is established between the behavior of Ti42Nb titanium alloys produced with building directions of 90${}^{\circ }$ and 45${}^{\circ }$ in order to extract additional information regarding the impact of loading conditions and anisotropy on mechanical behavior. The yielding strength coefficient $\sigma_{0.2}$ obtained for an axial strain ε of 0.2% is given in Figure 8. Whatever the building direction is, the yielding strength coefficient increases when the strain rate increases and/or the deformation temperature decreases. The former is a consequence of the movement of screw dislocations in BCC metals [49]. Indeed, high strain rate increases the dislocation velocity and requires higher applied stress for gliding of the screw dislocation segments. The latter is related to the fact that an increase in the deformation temperature induces an increase in the dislocations mobility. Additionally, regardless of the loading conditions, the Ti42Nb specimens produced with building direction of 45${}^{\circ }$ exhibit higher yield strength than those produced with a building direction of 90${}^{\circ }$.

In order to confirm the previous results concerning the effect of loading conditions on the thermomechanical behavior of the alloy, Vickers microhardness tests are carried out. Vickers microhardness measurements are shown in Figure 9 and Figure 10 for the Ti42Nb specimens manufactured with building directions of 90${}^{\circ }$ and 45${}^{\circ }$, respectively. In line with the obtained microhardness measurements, two points can be discussed: (i) a comparison between the microhardness of specimens after and before compression tests for a fixed building direction, and (ii) for a fixed loading conditions, comparing the Vickers hardness of deformed specimens produced with the building direction of 90${}^{\circ }$ to those manufactured with the building direction of 45${}^{\circ }$ to highlight the impact of anisotropic grain morphology.

On the one hand, at room temperature, the microhardness of the undeformed alloys (around 280 HV${}_{0.1}$) is lower than that of the deformed specimens for the two building directions. Therefore, an increase in the material microhardness is observed after compression tests at this temperature. These results reveal that the strain hardening governs the flow stress behavior of the Ti42Nb alloy. As a result, the decrease in the flow stress observed at room temperature can be related to the strain localization phenomena. A contribution of the increase in the pore size induced by severe plastic deformation to the decrease in the flow stress cannot be excluded. Similar observations are obtained by Wang et al. [26] for Ti40Nb alloy (in wt.%) fabricated by the LB-PBF process. On the other hand, when comparing the microhardness measurements obtained at different temperatures (673 K, 873 K and 1073 K) to those obtained at room temperature, the Vickers microhardness measurements drop progressively to ≈210 HV${}_{0.1}$ for both building directions. The results demonstrate that the mechanical behavior of the Ti42Nb alloy is controlled by strain softening for intermediate and high temperatures. At a fixed temperature, regardless of the building directions, a slight rise in the microhardness is obtained when the strain rate is increased from 1 to 10 s${}^{-1}$. Moreover, whatever the temperatures and strain rates are, the microhardness measurements illustrate higher values for specimens manufactured with a building direction of 45${}^{\circ }$ than those produced with a building direction of 90${}^{\circ }$. This result proves an accordance with the stress–strain curves as well as the strength coefficients. Vickers microhardness tests are carried out on the deformed sample in order to characterize un-molten Nb particles. According to the results, the microhardness of the un-molten Nb particles (around 240 HV0.1) is lower than that of the as-build material (around 280 HV0.1). Therefore, the un-molten Nb particles act as soft inclusions that have a negligible effect on the mechanical behavior of the alloy.

The strain rate sensitivity coefficient can be evaluated according to Arrhenius model to extract additional information regarding the impact of the loading conditions on the mechanical behavior. This coefficient *m* can be calculated at fixed axial strain ε and temperature *T* through the following equation [50,51]:(1)$$m={\left.\frac{\partial \log\sigma }{\partial \log\dot{\varepsilon }}\right|}_{\varepsilon ,T}$$
where σ and $\dot{\varepsilon }$ are, respectively, the axial stress and the axial strain rate. In the following, the strain-rate sensitivity coefficient is calculated at an axial strain of 10%. Figure 11 shows the evolution of the strain-rate sensitivity coefficient as a function of temperature. Based on the results, the as-LB-PBFed Ti42Nb alloy manufactured with a building direction of 90${}^{\circ }$ shows a moderate progression of the strain-rate sensitivity coefficient for temperature lower than 673 K and a significant increase for higher temperatures. For the Ti42Nb specimens produced with a building direction of 45${}^{\circ }$, a progressive increase in the strain-rate sensitive coefficient according to the temperature is obtained.

### 3.4. Metallographic Analysis

Ti42Nb parts produced in situ by laser-based powder bed fusion have poor surface integrity. Therefore, Ti42Nb implants need to be machined in order to obtain functional surface finish. The machining process almost leads to subjecting the parts to high temperatures and high strain rates. Therefore, it is highly important to understand the effect of these loading conditions on the thermomechanical response of Ti42Nb alloy and to obtain a stress–stress curve that allows us to identify constitutive models used to simulate micromachining.

During the compression tests, Ti42Nb alloy specimens experience severe loading conditions such as large deformations and high strain rates. For β titanium alloys, these loading conditions can induce some microstructure alterations, such as stress-induced martensitic transformation. To gain insight into the presence of stress-induced martensitic transformation after compression tests, X-ray diffraction analysis is systemically performed on the deformed specimens. For illustration, the XRD patterns of Ti42Nb specimens manufactured with a building direction of 90${}^{\circ }$ are shown in Figure 12. Identical results are obtained for Ti42Nb specimens manufactured with a building direction of 45${}^{\circ }$. According to the results, whatever the loading conditions are, XRD patterns illustrate that the deformed specimens exhibit only single β-phase microstructure. The stress-induced martensitic transformation is therefore not observed in this work.

Based on metallographic observations, different microstructure alterations are observed after the compression tests. On the one hand, at low deformation temperatures, a localization of the plastic deformation, which induces the formation of the adiabatic shear band (ASB), is observed whatever the building direction is. Temperatures and strain rates leading to the formation of ASBs are summarized in Table 2.

As shown in Figure 13 and Figure 14, when the conditions are met for shear banding, ASBs are oriented with an angle of ±45${}^{\circ }$ with respect to the compression direction. At room temperature, adiabatic shear bands are systematically formed. However, the formation of ASBs depends on the applied strain rates when the specimens are deformed at high temperatures. When the adiabatic shear bands are not observed, highly deformed regions are formed in the central region of the compression specimens. The concentration of the plastic deformation induces a strain localization mode perpendicular to the compression direction. In this study, the formation of the highly deformed regions during uniaxial compression trials can contribute to strain softening.

In the high temperature range, the behavior of the Ti42Nb alloy produced with both building directions is influenced by dynamic recrystallization. As illustrated in Figure 15, EBSD analyses of the deformed specimens show recrystallized grains preferably formed at triple junctions and along grain boundaries. For both building directions, the loading conditions that lead to the occurrence of dynamic recrystallization are listed in Table 3. For these conditions, a steady-state flow stress behavior is systematically detected. As suggested by [52,53], the steady-state flow stress can be related to dynamic recrystallization. According to Figure 15, a strong change of the crystallographic orientation of deformed Ti42Nb alloy is observed. As shown in the pole figures (Figure 15c), the feature of the fiber texture of the cube components ({100}〈101〉) appears. Compared to the pole figures of the un-deformed material, it should be noted that the intensity of fiber texture of the deformed material changed. For the undeformed material, the maximum value of the intensity of 〈100〉 pole is about 4.82. However, the maximum intensity value of the latter is 7.35. This texture change can be explained by the localization of the plastic deformation perpendicular to the compression direction. This change of the crystallographic orientation can contribute to the strain softening at this temperature range.

## 4. Conclusions

The goals of this study are to investigate the impact of the as-build LB-PBF directions and the loading conditions (i.e., temperatures and strain rates) on the mechanical behavior of Ti42Nb titanium alloy. In this context, cylindrical specimens are produced in situ by an additive manufacturing process on Ti and Nb elemental powders and then deformed. The uniaxial compression tests are performed at two strain rates, 1 and 10 s${}^{-1}$, and temperatures ranging from 298 to 1073 K. Two different building directions are considered. The main conclusions that could be drawn are listed below:•The as-LB-PBFed Ti42Nb microstructure is only composed of β-phase. The *α*″ martensitic phase, which is possibly developed during LB-PBF process, is not observed.•For both building directions, the Ti42Nb microstructures show the presence of un-melted Nb powders.•Whatever the loading conditions are, the flow stress of the specimens produced with a building direction of 45${}^{\circ }$ is higher than that of the specimens fabricated with a building direction of 90${}^{\circ }$.•Due to the low strain-rate sensitivity, deformed Ti42Nb specimens are subjected to adiabatic shear band at low temperature and high strain rate. In contrast, when conditions are not met for shear banding, the plastic deformation is localized in the central region of the specimens perpendicular to the compression direction.•The stress–strain curves obtained from these experimental datasets are used to identify the material parameters of a crystal plasticity-based constitutive model over a wide range of temperatures.

## Figures and Tables

**Figure 1 materials-15-00509-f001:**
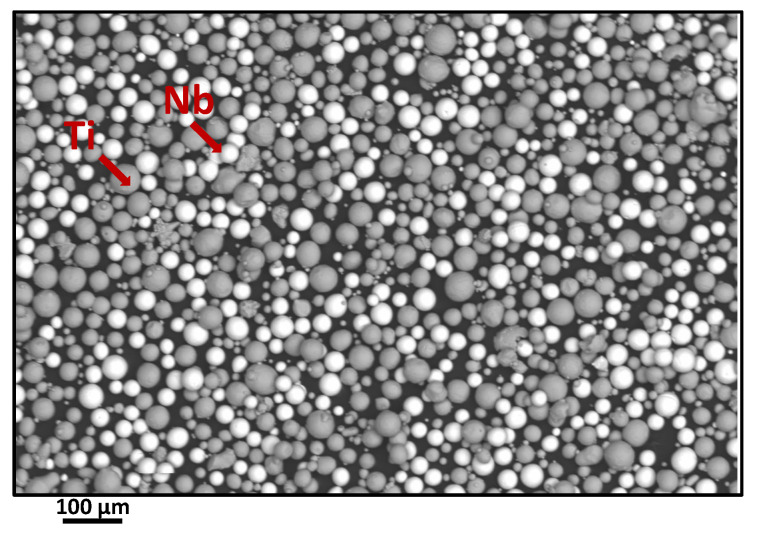
Scanning electron microscopy (SEM) observation showing the powder morphologies.

**Figure 2 materials-15-00509-f002:**
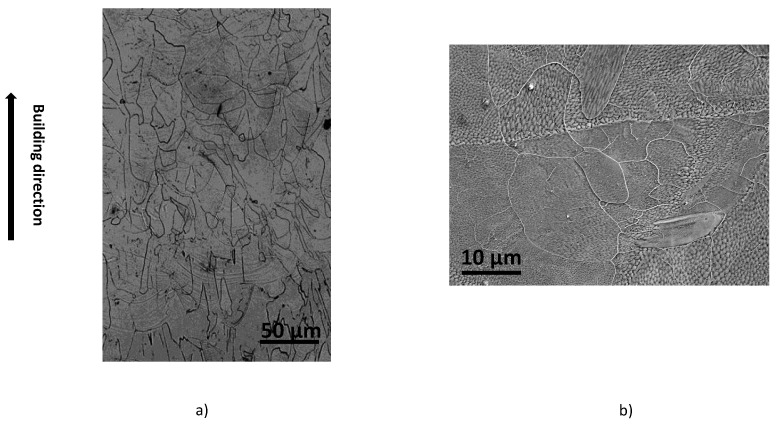
OM (**a**) and SEM (**b**) observations showing the single β microstructure of the as-LB-PBFed Ti42Nb alloy manufactured at 90${}^{\circ }$.

**Figure 3 materials-15-00509-f003:**
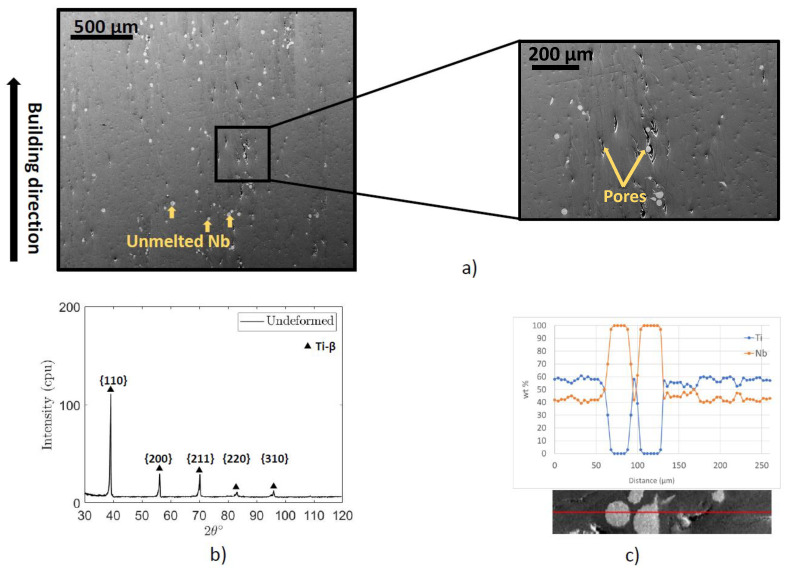
(**a**) Scanning electron microscopy (SEM) observations showing the as-LB-PBFed Ti42Nb microstructure, the undissolved Nb particles and the pores, (**b**) X-ray diffraction pattern, and (**c**) EDX quantification near a niobium nodule.

**Figure 4 materials-15-00509-f004:**
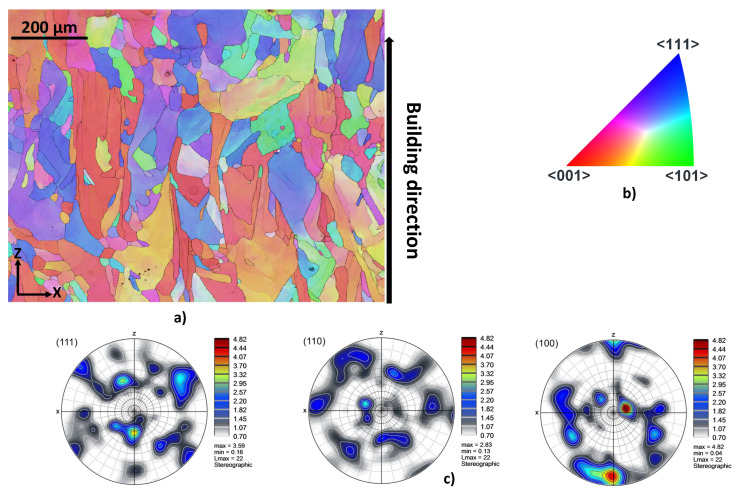
EBSD map taken in the center of the as-LB-PBFed Ti42Nb specimens produced with a building directions of 90${}^{\circ }$ (projection axes $\langle 100\rangle$): (**a**) inverse pole figures (IPF (Z)), (**b**) IPF coloring legend, (**c**) pole figures (PF).

**Figure 5 materials-15-00509-f005:**
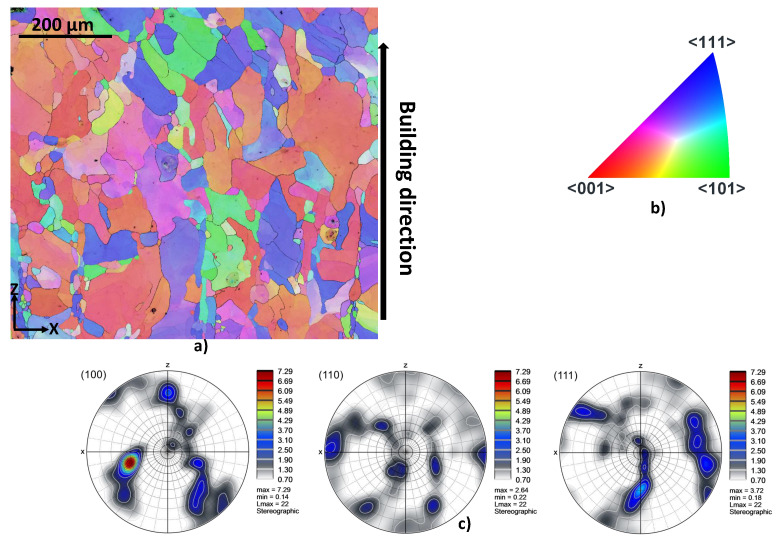
EBSD map taken in the center of the as-LB-PBFed Ti42Nb specimen produced with a building directions of 45${}^{\circ }$ (projection axes $\langle 101\rangle$): (**a**) inverse pole figures (IPF (Z)), (**b**) IPF coloring legend, (**c**) pole figures (PF).

**Figure 6 materials-15-00509-f006:**
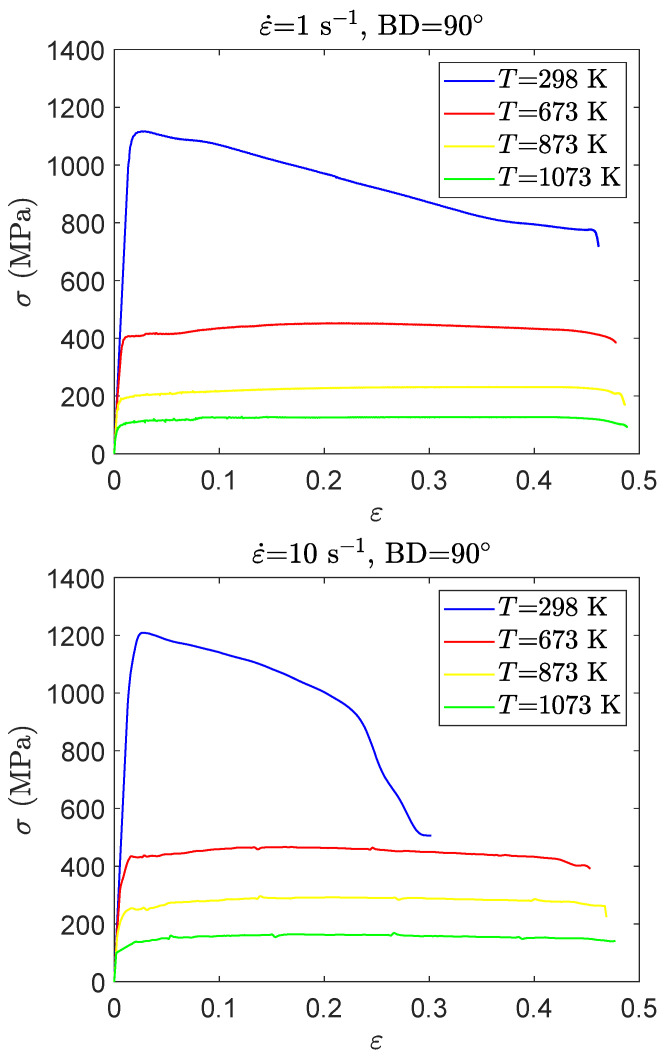
Stress–strain curves for the compression tests carried out on Ti42Nb specimens manufactured with a building direction of BD-90${}^{\circ }$ deformed at different temperatures and strain rates.

**Figure 7 materials-15-00509-f007:**
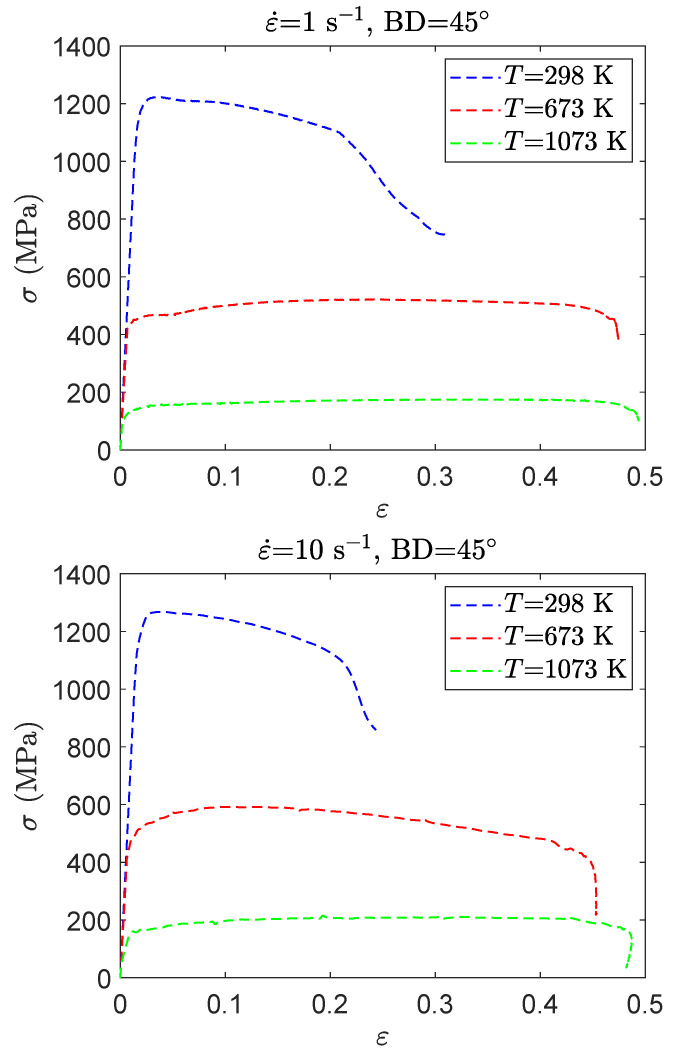
Stress–strain curves for the compression tests carried out at different temperatures (298 K, 673 K, and 1073 K, as well as two strain rates (1 s${}^{-1}$ and 10 s${}^{-1}$), on Ti42Nb titanium alloy produced with a building direction of 45${}^{\circ }$.

**Figure 8 materials-15-00509-f008:**
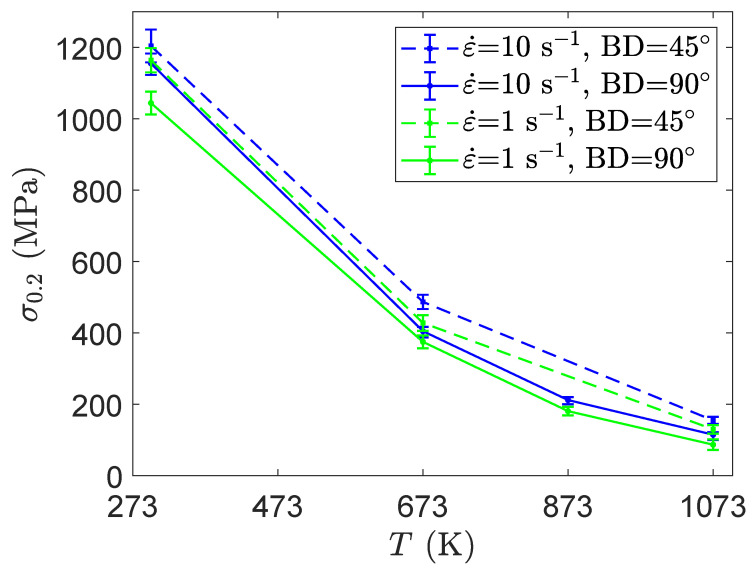
Variation of the strength coefficient $\sigma_{0.2}$ for the Ti42Nb alloy produced with two building directions as a function of temperature.

**Figure 9 materials-15-00509-f009:**
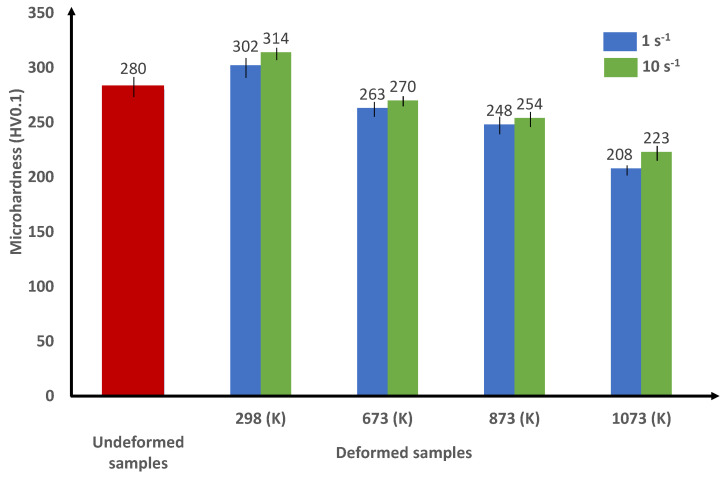
Vickers microhardness of the un-deformed and the deformed Ti42Nb alloy produced by LB-PBF with BD-90${}^{\circ }$ at different temperatures and strain rates.

**Figure 10 materials-15-00509-f010:**
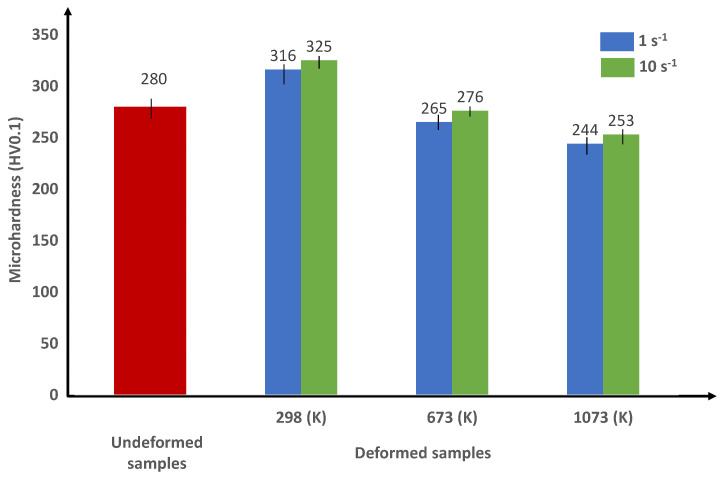
Vickers microhardness of the un-deformed and the deformed Ti42Nb alloy produced by LB-PBF with BD-45${}^{\circ }$ at different temperatures and strain rates.

**Figure 11 materials-15-00509-f011:**
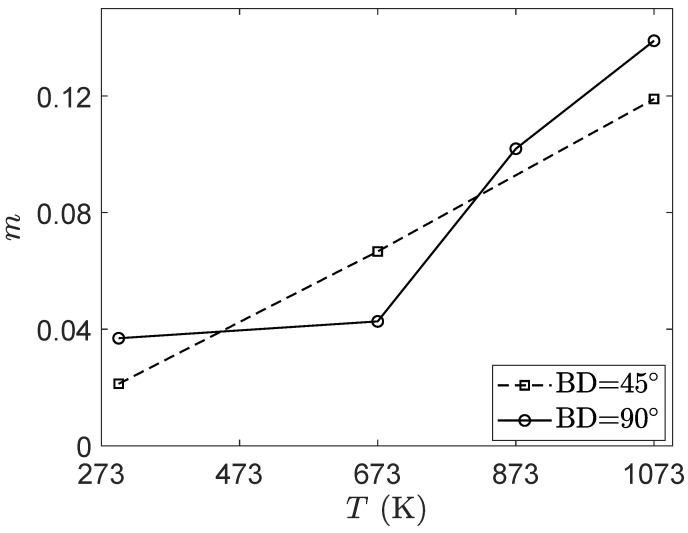
Variation of the strain-rate sensitivity coefficient *m* according to the temperature.

**Figure 12 materials-15-00509-f012:**
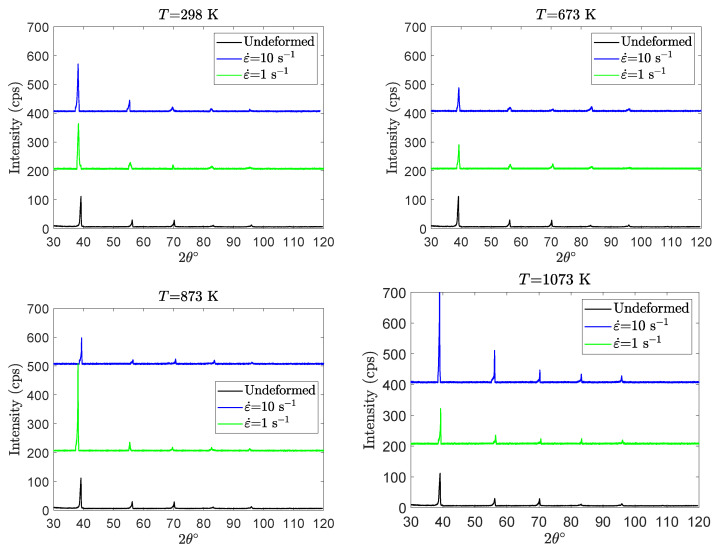
XRD patterns obtained with a Cu Kα radiation for the compression tests carried out on Ti42Nb alloy produced with 90${}^{\circ }$ building direction at temperatures ranging from 298 to 1073 K and two strain rates: 1 s${}^{-1}$ and 10 s${}^{-1}$.

**Figure 13 materials-15-00509-f013:**
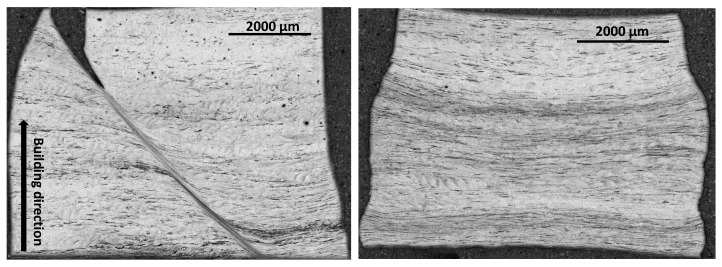
OM observations of Ti42Nb specimens manufactured with a building direction of 90${}^{\circ }$ deformed at a strain rate of 10 s${}^{-1}$ and two temperatures: (**left**) 298 K and (**right**) 1073 K.

**Figure 14 materials-15-00509-f014:**
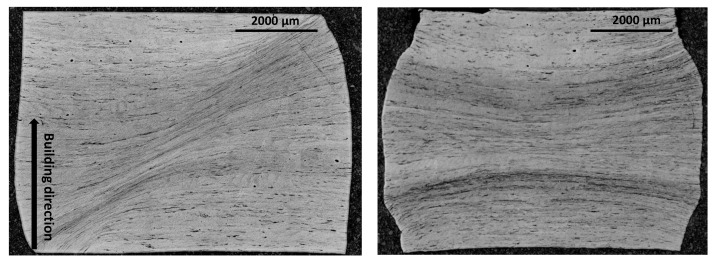
OM observations of Ti42Nb specimens manufactured with a building direction of 45${}^{\circ }$ deformed at a strain rate of 10 s${}^{-1}$ and two temperatures: (**left**) 298 K and (**right**) 1073 K.

**Figure 15 materials-15-00509-f015:**
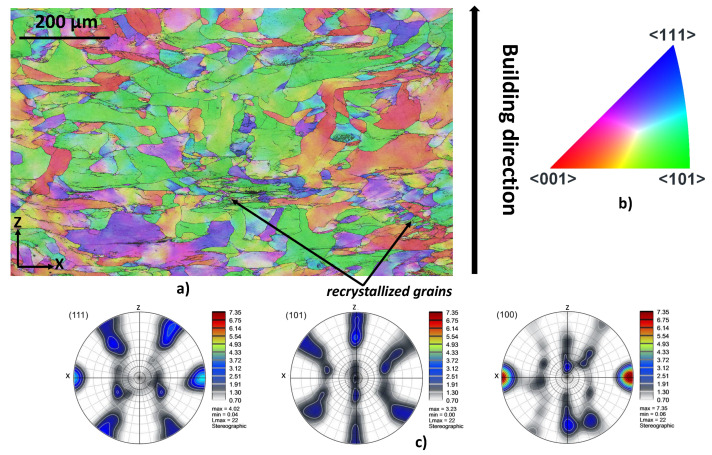
EBSD map performed in the center of Ti42Nb specimen (Figure 13 right) produced with a building direction of 90${}^{\circ }$ and deformed at a temperature of 1073 K and a strain rate of 10 s${}^{-1}$. The compression direction is parallel to the Z-axes. (**a**) Inverse pole figures (IPF(Z)), (**b**) IPF coloring legend, (**c**) pole figures (PF).

**Table 1 materials-15-00509-t001:** Chemical compositions (in wt %) of the powders and the as-build alloy.

Element	Ti	Nb	C	S	H	N	O
Mixed powder	57.29	42.5	0.029	0.007	0.005	0.007	0.18
As-built alloy	58.1	41.6			0.01	0.03	0.22

**Table 2 materials-15-00509-t002:** Summary of the occurrence of adiabatic shear band (•) and highly deformed region (∘) during uniaxial compression of the Ti42Nb alloy manufactured with building direction of 90${}^{\circ }$ and 45${}^{\circ }$.

	LB-PBF Building Directions			
	90°		45°	
*T* (K)	Strain Rate (s^−1^)			
	1	10	1	10
298	•	•	•	•
673	∘	∘	∘	•
873	∘	∘		
1073	∘	∘	∘	∘

**Table 3 materials-15-00509-t003:** Summary of the occurrence of dynamic recrystallization during uniaxial compression of the Ti42Nb alloy produced at BD-0${}^{\circ }$ and BD-45${}^{\circ }$.

	BD-90${}^{\circ }$		BD-45${}^{\circ }$	
*T* (K)	Strain Rate (s^−1^)			
	1	10	1	10
298				
673				
873	•	•		
1073	•	•	•	•

## Data Availability

All the data is available within the manuscript.

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
