# Peer review of "Impact of the Loading Conditions and the Building Directions on the Mechanical Behavior of Biomedical β-Titanium Alloy Produced In Situ by Laser-Based Powder Bed Fusion"

_materials, 2022, doi:10.3390/ma15020509_

Round 1

Reviewer 1 Report

Manuscript with number “materials-1477611” was reviewed, flowing comments are suggesting:

Please use the standard terminologies of ASTM/ISO 52900 for correct terminologies of the Additive Manufacturing process. Use laser based-powder bed fusion (LB-PBF) instead of selective laser melting (SLM).

The introduction part is too long, please summarize it.

It is also suggested to add novel paper in the introduction, following paper suggested:

Titanium fabricated by selective laser melting: microstructure, wear and corrosion behavior in different orientations

A review on the performance characteristics, applications, challenges and possible solutions in electron beam melted Ti-based orthopaedic and orthodontic implants

Design, additive manufacture and clinical application of a patient-specific titanium implant to anatomically reconstruct a large chest wall defect

Supportless printing of lattice structures by metal fused filament fabrication (MF3) of Ti-6Al-4V: Design and analysis

A study on surface morphology and tension in laser powder bed fusion of Ti-6Al-4V

Please double-check the numbering of headings.

Results have been just reported, please add the mechanism and reason for your findings.

 Please add full statistical analysis related to results, it is also suggested to present the results in a table.

Reviewer 2 Report

The manuscript materials-14776611 “Impact of the loading conditions and the building directions on the mechanical behavior of biomedical β-titanium alloy produced in-situ by selective laser melting” presents results related to compression test at various temperatures of a Ti-Nb alloy produced by in-situ alloying of elemental powder blends using selective laser melting.

In the present form the paper is not clearly organized. The section “2. Materials and Methods” (in the manuscript the numbering must be corrected) should not contain results. E.g. the micrographs in Figure 2 -5 belong to the results section.

The paper appears quite long for the content and some figures are very large (e.g. Fig 1.) while other figures contain much white space and should be reorganized / compressed (Fig 3, Fig 6). Personally, I would recommend to remove Fig 1, Fig 6 and Fig 7 from the manuscript as they are purely illustrative.

There are a couple of typing and grammar errors such as “mon-columnar” and “…which is may induced..:” on page 4. The authors should carefully check the manuscript again and correct these mistakes.

A couple of clarifications with regard to the content are required:

  1. Chemical composition of powder blend is given (table 1). But what is the composition of the as-build materials.
  2. What is meant by projection axes in the figure captions of Fig 4 and 5
  3. Explain why 45° samples exhibit higher flow stresses than 90° samples.
  4. Why is σε0.5% used instead of the typical σ0.2?
  5. Is it the assumption really realistic: “…an increase of the mean free path of dislocations due to the dissolution of un-melted Nb powder during heating of the specimens from room temperature to the test temperature”. This assumption should be reconsidered.
  6. The XRD results shown in Fig. 14 only show that the material consists mainly of bcc-Ti for all conditions. Peak broadening can not be recognized (because a magnified comparison of single reflections would be required). Is it also the case for the samples deformed at high temperatures? A figure showing e.g. the full width at half maximum (FWHM) would be much more helpful than the whole diffractograms.
  7. Can recrystallization also be proved by XRD? Can you highlight dynamic recrystallization in Fig 17 showing a magnified section.
  8. Why are these shear bands “adiabatic”? Are there any proofs for this assumption?
  9. For medical application certainly in-situ alloying is not a feasible production method. Comment on how the research will be continued.
  10. What is the impact of the unmolten Nb particles on the mechanical behaviour?
  11. After reading the manuscript it remains unclear why these tests are relevant for biomedical applications. Please comment on that in the manuscript.

Reviewer 3 Report

The work investigates deformation mechanisms of LPBF Ti42Nb alloy. The paper structure requires major revision as many results appear in the methods section. Furthermore, some of the results are completely missing associated methods sections. The data investigating the compression performance of 45/90 orientation samples at different temperatures is relevant but not well discussed in terms of overall applications. Despite providing new data for the field, no insightful conclusions are made about the use of LPBF Ti42Nb for biomedical applications. The work requires major revision before publication. Further improvements are detailed below;

  1. Introduction: English quality requires revision
  2. Materials and Methods
    1. A lot of results are included in the methods. They should be moved to results and discussion
    2. No indication of etchant used to view microstructure
    3. It is mentioned that the measured lattice parameter is consistent with that for BCC beta phase Ti42Nb alloy but no reference is given
    4. The ultrasonic modulus and XRD data has no related methods section
    5. Tomography analysis is mentioned with no related methods section
    6. To show preferred orientation in EBSD, please include the multiple of uniform density (MUD) value
    7. Please give more indication as to how the deformed specimens were prepared for EBSD analysis
  3. Results and discussion
    1. Microhardness methods is included in results. Move to methods section
    2. Discuss the mechanical observations in light of applications for biomedical implants

Reviewer 4 Report

This paper is overall excellent and well organized. However, I have a question, as well know that the Additive Manufacturing (laser powder bed fusion) is a near net shape forming technology, and the final part can be obtained by this method, and thus I'm confused that why you do the hot compression tests and study the effect of temperatures and strain rates on the deformation behavior of Ti-Nb alloys produced by laser powder bed fusion. In other words, the motivation of this research is not clear, so I suggest you rewrite introduction section and make it clear that why you do  this research that the effect of temperatures and strain rates on the hot deformation behavior. 

Round 2

Reviewer 1 Report

The authors have addressed the issues raised previously, and the manuscript is suitable for publication in its current form.

Author Response

We would like to thank the reviewer for their thorough reading of the
manuscript. 

Reviewer 2 Report

The paper materials-1477611 has been renamed to “Impact of the loading conditions and the building directions on the mechanical behaviour of biomedical β-titanium alloy produced in-situ by laser based-powder bed fusion”.

Please note that it should be “…laser-based powder fusion”. The abbreviation LB-PBF is, by the way, already used in the abstract without being defined, and the mistake (wrong position of hyphen) is repeated several times in the introduction part.

The paper, especially the structure, has been improved but it is still not mature.

  • While it was given in the coverletter, the table for the composition of the as-build material is not given in the manuscript. It should be added to table 1
  • In Fig 5a the arrows pointing to the pores are shifted. This has to be corrected.
  • There is confusion in the manuscript related to the IPF maps. The response in the letter is related to pole figures. While in the original manuscript no pole figures but IPF maps were given, in the new version pole figures are added. The three pole figures are equivalent, and the two superfluous should be removed (or (110) and (111) should be added. On the other hand, it is still not clear what projection axis means related to the inverse pole figure maps. (here it should be added to the figure caption which sample direction is related to the colour coded IPF map. There is no description related to the added pole figures (Figures 6c and 7c). Please check whether the directions x,y are correct in the pole figures. Also the description of the EBSD result in the text has to be improved. Please also edit Fig 17 the same way, so that a comparison of as-fabricated and deformed is possible.
  • The XRD results presented in Fig 14 are still more or less meaningless as it is not possible to see the details (peak broadening). Better show details or remove the figure.
  • There are some explanations in the cover letter that, however, are not addressed in the paper (remark 4, remark 6, remark 9, remark 12, remark 13).
  • The samples are heated with 10K/s + 30s holding time before deformation. This means that <2min for the 800°C experiments. In this short time according to own experiments no significant dissolution of Nb particles can be expected. That was the reason for remark 8.

Author Response

We would like to thank the reviewers for their thorough reading of the
manuscript. We believe we have addressed all comments in the revised version. See the responses below for details. Responses are highlighted with an
italic font.

Reviewer 3 Report

The manuscript has been significantly improved upon review. As micro-machining has been used as a justification for investigating high temperatures and strain rates, please add a final paragraph in section 3.4 and a conclusion point suggesting how the observed results should guide micro-machining processes for Ti42Nb implants. This will help to strengthen the applicability of the study. 

Author Response

Following the above suggestion and to highlight the originality of this investigation, section 3.4 (Metallographic analysis) and section 4 (Conclusions) have been modified as follows:

"... Ti42Nb parts produced in-situ by laser-based powder bed fusion have
poor surface integrity. Therefore, Ti42Nb implants need to be machined in
order to obtain a functional surface finish. The machining process almost leads
to subjecting the parts to high temperatures and high strain rates. Therefore,
it is highly important to understand the effect of these loading conditions on
the thermomechanical response of Ti42Nb alloy and to obtain stress-stress
curves that allow identifying constitutive models used to simulate micromachining...

... The stress-strain curves obtained from this experimental dataset are
used to identify the material parameters of a crystal plasticity-based constitutive model over a wide range of temperatures."

Round 3

Reviewer 2 Report

Before publication of the manuscript materials-1477611 “Impact of the loading conditions and the building directions on the mechanical behaviour of biomedical β-titanium alloy produced in-situ by laser-based powder bed fusion” I would like to suggest the following improvements:

  • My remark to the abbreviation LB-PBF was misunderstood. It was not my intention to change the abbreviation. Please stay with LB-PBF. Only the position of the hyphen in “laser-based powder bed fusion” was addressed.
  • A description of the subfigures should be added to the figure caption. This has already been addressed in my previous review. This applies to Fig.2, Fig.4, Fig.5, Fig15.
  • 4 and Fig 5: According to the explanation in the cover letter right, the IPF map in Fig. 4 is for the z-direction of the sample. This is reasonable and what I expected. To make this clearer for the reader I recommend instead or in addition to the vector notation <100> to write IPF for z-direction of sample (or something like that), because the sample´s coordinate system is described that way. I wonder why a different direction was used for the IPF map in Fig 5. (projection axis <101>). Why not used IPF in z-direction as in Fig. 4? Because a texture 45° inclined to the build direction is expected?
  • In the caption of Fig.15 it should be: “..Ti42Nb specimen (figure 13 right)…”
  • In Fig 15 thanks to the pole figures a nice texture of the deformed specimen is shown. Please explain the texture.

Author Response

  • We would like to thank the reviewer 2 for his thorough reading of the
    manuscript. We believe we have addressed all comments in the revised version. See the responses below for details. Responses are highlighted with an italic font.
